# Capillary Refill Time and Serum Lactate as Predictors of Mortality and Postoperative Extracorporeal Membrane Oxygenation Requirement in Congenital Heart Surgery

**DOI:** 10.3390/children10050875

**Published:** 2023-05-13

**Authors:** Gustavo Cruz, Santiago Pedroza Gómez, Akemi Arango, Paula A. Guevara, Carlos González, Jesus Aguirre, Andrea Valencia-Orozco, Antonio J. Suguimoto

**Affiliations:** 1Departamento de Anestesiología, Fundación Valle del Lili, Cra 98 No. 18–49, Cali 760032, Colombia; 2Centro de Investigaciones Clínicas, Fundación Valle del Lili, Cra 98 No. 18–49, Cali 760032, Colombia; santiago.pedroza@fvl.org.co (S.P.G.);; 3Universidad Icesi, Facultad de Medicina, Departamento de Anestesiología, Calle 18 No. 122–135, Cali 760031, Colombia; 4Departamento de Pediatría, Fundación Valle del Lili, Cra 98 No. 18–49, Cali 760032, Colombia

**Keywords:** congenital heart surgery, perfusion biomarkers, capillary refill time, seric lactate, mortality, extracorporeal oxygenation

## Abstract

Multiple tissue perfusion markers are described to guide therapy in critically ill pediatric patients undergoing congenital heart surgery. Given the advantages of capillary refill time, our goal is to determine its predictive capacity for mortality and postoperative extracorporeal oxygenation requirements in congenital heart surgery and compare it to serum lactate. We conducted a prospective cohort observational study in a single high-complexity university hospital. Serum lactate and capillary refill time were measured at five predetermined time points: preoperative, immediate postoperative, 6, 12, and 24 h after the surgery. Prolonged immediate postoperative, 6 h, and 12 h capillary refill time measurements turned out to be independent risk factors for both outcomes. The capillary refill time area under the curve ranged between 0.70 and 0.80, while the serum lactate resulted between 0.79 and 0.92 for both outcomes. Both tissue perfusion markers resulted in mortality and extracorporeal oxygenation requirement predictors. Given the advantages of capillary refill time over serum lactate, a monitoring strategy including these two perfusion markers should be considered for congenital heart surgeries.

## 1. Introduction

The assessment and monitoring of tissue perfusion is mandatory in congenital heart surgeries (CHS) [1]. Multiple tissue perfusion biomarkers are described to guide therapy in critically ill pediatric patients [2,3,4]. Among these, serum lactate (SL) is considered the gold standard [5]; however, it has important disadvantages such as the need for blood samples, which carries the risk of anemia and infection. In addition, SL testing may not be widely available in all hospitals or clinics, especially in low-income medical settings [6]. On the other hand, additional mechanisms of hyperlactatemia related to metabolic stress (exercise, recent surgery, and hospitalization) and delayed clearance (liver disease, ongoing dysoxia) can lead to erroneous interpretations [2,7].

Central venous oxygen saturation, another widely used biomarker, requires serial measurements with invasive devices and is unreliable in residual intracardiac shunts [7,8]. Capillary refill time (CRT) has been used as an alternative measure to assess tissue perfusion [9,10]. The skin, an organ with no autoregulation mechanisms for blood flow, provides us with valuable information on the state of perfusion [9]. This measurement can be monitored in real time with no devices, at the patient’s bedside, does not require serial blood samples, and generates no additional costs to the health system [11].

The use of CRT has been mainly studied in adult patients [9,12,13,14,15]. This clinical marker has been shown to be an adequate predictor of multiple organ dysfunction and mortality in septic shock [12,13]. In pediatric patients, Tibby et al. studied CRT in children undergoing CHS but found no statistical relationship with mortality [10]. There is currently no conclusive evidence in the literature regarding CRT monitoring in CHS, and no study was found in this population that compared different methods of tissue perfusion biomarkers and clinical markers. Taking advantage of this knowledge gap, a prospective observational study was conducted on pediatric patients undergoing CHS to assess the predictive capacity of CRT and SL for mortality and postoperative extracorporeal membrane oxygenation (ECMO) requirement. We predict that CRT is a useful tissue perfusion clinical marker for predicting both outcomes (mortality and ECMO) and, consequently, for monitoring postoperative patients in the studied population.

## 2. Materials and Methods

### 2.1. Study Design

A prospective observational study was conducted in la Fundación Valle Del Lili in Cali, Colombia. All pediatric patients (under 5 years of age) undergoing CHS between November 2020 and May 2022 were included. Patients with any oncoproliferative disease or low-complexity heart surgeries with no perioperative extracorporeal circulation requirement were excluded.

### 2.2. Variables and Measurements

Initially, the medical history was queried for patient demographics and pre-operative clinical status. We identified the age, sex, weight, height, surgery condition, medical history, preoperative laboratories, and risk adjustment for CHS (RACHS-1). SL and CRT were measured at predetermined time points; preoperative (PO), immediate postoperative (IPO) (on admission to the pediatric ICU), at 6, 12, and 24 h after the surgery. Each patient was assigned a study ID to ensure anonymity. The data corresponding to the outcomes in the first 24 h were collected by the critical care physician on duty in the case report forms (CRF) of the study. The outcomes after 30 days were obtained by reviewing the clinical history of the subjects who remained hospitalized. The collected CRF were transcribed in a clinical database management system of the institution by the researchers, a platform for protected data collection.

All CRT measurements were performed by 2 individuals (the cardiovascular anesthesiologist in charge of the case and the critical care physician from the pediatric cardiovascular ICU) after standardizing the measurement technique. Twenty-five percent of the measurements were corroborated by a researcher, obtaining a 100% concordance. The measurement was performed by applying firm pressure to the right heel with a glass (see Appendix A). If not possible, it was performed on the ventral surface of the distal right index finger. Firm pressure was applied for 5 s, and the CRT was measured with a stopwatch. The patient’s temperature had to be >35.9 °C and a temperature in the ICU cubicle > 22 °C. The evaluated outcomes were 30-day mortality and the postoperative requirement of ECMO.

#### 2.2.1. Anesthetics, Inotropes, Vasopressors, and Transfusion Strategy

For induction of anesthesia, a combination of sevoflurane (minimum alveolar concentration = 0.5–1%) and either dexmedetomidine (0.3–0.8 mcg/kg/h) or fentanyl (1–3 mcg/kg/h) was used. For neuromuscular blockage, rocuronium was administered in a dose range of 0.6–1.2 mg/kg. Mechanical ventilation was performed with a tidal volume of 6–8 mL/kg and a ventilatory frequency aimed at achieving an end-tidal carbon dioxide level between 35 and 45 mmHg.

Blood reservation included one unit of fresh frozen plasma plus one unit of packed red blood cells (RBC) or fresh whole blood for purging the extracorporeal circulation machine (approximately 150 mL). From this blood, before the start of ECMO, one or two 60 mL syringes were provided for exclusive use by the cardiovascular anesthesiologist. In RACHS-1 ≥ 4 surgeries one unit of plateletpheresis (approximately 150 mL) and 4 units of cryoprecipitate were requested. Platelets were administered to all children under 8 kg (kg) undergoing RACHS-1 ≥ 4 surgeries (10 cc/kg). Cryoprecipitate was given to all children under 6 months undergoing RACHS-1 ≥ 4 surgeries (10 cc/kg) to achieve a fibrinogen level greater than 200 mg/dL. To avoid fluid overload, platelets and cryoprecipitate were administered while ultrafiltration was performed (approximately 15 min). Prothrombin Complex Concentrate (octaplex and blood clotting factors II-VII-IX-X) was available for massive bleeding, aorta surgeries, or at the discretion of the anesthesiologist.

In RACHS-1 ≥ 4 surgeries cell saver was performed. The recovered cells were taken to the ICU but expired 4 h after being processed. The cell salvage removed 92% of the heparin and potassium and had an average hematocrit of 60%.

The administration of inotropes and vasoactive medications was based on hemodynamic variables, echocardiographic monitoring, and perfusion monitoring (arterial pressure, heart rate, blood oxygen saturation, capnography, ventricular function, valvular function, flow velocity in pulmonary veins, CRP, SL, mixed venous oxygen saturation, and cerebral and somatic near-infrared spectroscopy). For surgeries with RACHS-1 ≥ 3, milrinone was administered at a dose of 0.4–1.2 mcg/kg/min. The vasoactive medications we used were epinephrine at a dose of 0.03–0.2 mcg/kg/min, vasopressin at a dose of 0.002 UI/kg/min (preferred in patients with pulmonary hypertension), and norepinephrine at a dose of 0.05–0.2 mcg/kg/min.

#### 2.2.2. Statistical Method

A descriptive analysis was made. The normality of continuous variables was analyzed using the Shapiro–Wilk test and a box plot. If the assumption of normality was rejected, data were reported as median (interquartile range (IQR)). Categorical variables were expressed in number and percentage. The receiver operating characteristic (ROC) curve was used to assess the ability of different SL levels and CRT to predict 30-day mortality and the postoperative requirement of ECMO. To identify the cut-off points (COP), which refers to the value closest to the point of maximum sensitivity and specificity, the Youden J index was calculated as J = sensitivity + (specificity − 1). Sensitivity (S), specificity (Sp), and predictive values (PPV, NPV) were calculated for each measurement time. A problem that appears frequently when applying logistic regression to small data sets is a partial or total separation, which results in the non-existence of maximum likelihood estimators [14]. Logistic regression with Firth’s penalty was chosen, with the aim of predicting 30-day mortality and the requirement of ECMO based on the available markers. A univariate and multivariate regression analysis was performed including SL and CRT for each measurement time. Significance was accepted at *p* < 0.05 for all tests. Analyzes were carried out in R V.4.1.1 (R Foundation for Statistical Computing) through RStudio V.1.4.1717.

Ethical considerations: This study was approved by the Institutional Ethics Committee (approval number 365-2020 Act No. 29 of 4 November 2020) following national and international recommendations for human research. In accordance with resolution 8430 of 1993, this study was considered risk-free, and the waiver of informed consent was requested and obtained.

## 3. Results

Six hundred measurements of SL and CRT of 120 pediatric patients undergoing CHS were analyzed. The median age of the participants was 4.5 months (0.8, 11.0) and 50% were male. The most common age group was infants (54.2%), followed by newborns (25%) and toddlers (20.8%). Thirty-five percent of the patients had low weight-for-height and 22.5% were preterm (birth < 37 weeks). Regarding the outcomes evaluated, 12 (10%) patients died and 16 (13.3%) required ECMO after surgery (Table 1). Mortality and postoperative ECMO requirement occurred mostly in RACHS-1 5-6 surgeries (Table 2). The median and interquartile range of CRP maintained similar values through all measurement times. In contrast, the values in SL varied, decreasing over time (Table 3).

In the univariate analysis, SL and CRT were predictors of both 30-day mortality and the ECMO requirement. For mortality, the CRT obtained an odds ratio (OR) between 1.57 and 4.31 with a greater predictive capacity at 6 h (OR 2.98, Confidence Interval (CI) 1.54–6.27, *p* = 0.001) and at 12 h (OR 4.31, CI 1.98–11.19, *p* = 0.000). The OR of SL ranged between 1.30 and 1.51, being more significant at 6 h (OR 1.51, CI 1.26–1.89, *p* = 0.000). For the ECMO requirement, the CRT obtained an OR between 1.63 and 3.03 with a greater predictive capacity at 6 h (OR 2.74, CI 1.56–5.16, *p* = 0.000) and at 12 h (3.03, CI 1.65–6.13, *p* = 0.000). The OR of lactate ranged between 1.22 and 2.04, being more significant at 24 h (OR 2.04, CI 1.42–3.26, *p* = 0.000). Both perfusion markers at the different measurement times were statistically significant (*p* values < 0.05) (Table 4 and Table 5).

In the multivariate analysis for mortality, PO, 6 h and 24 h SL, and IPO, 6 h and 12 h CRT maintained statistical significance (Table 4). Regarding the requirement of postoperative ECMO, IPO, 6 h, 12 h, and 24 h CRT, and PO, 6 h and 24 h SL maintained statistical significance (Table 5).

Overall, SL obtained a higher area under the curve (AUC) than CRT for both outcomes (Figure 1, Figure 2, Figure 3 and Figure 4). The AUC of CRT for mortality and ECMO ranged between 0.7 and 0.8 in all measurement times. The COP varied between 3 and 4 s. SL maintained an AUC between 0.85 and 0.95 for mortality in all measurement times and the COP ranged between 1.5 and 3.2 mmol/L. For the ECMO requirement, the AUC ranged between 0.79 and 0.9, obtaining COP from 1.7 to 5 mmol/L (Table 6 and Table 7).

## 4. Discussion

This observational study in pediatric patients undergoing CHS showed that both CRT and SL were associated with 30-day mortality and the need for postoperative ECMO. IPO, 6 h, and 12 h CRT turned out to be independent risk factors for both outcomes, and at 24 h for postoperative ECMO requirement. In addition, the AUC of CRT showed an acceptable discriminative capacity for the two outcomes, with better performance at IPO and 12 h for mortality. Similarly, Shaker et al. found better predictive values of CRT at 6 and 12 h than in other measurement times in patients with abdominal sepsis [5] and Morocho et al. reported a CRT at 6 h as an independent risk factor for mortality in septic shock [12].

We used the Youden index to identify the optimal COP point at the different measurement times. Other studies have reported similar optimal COP found in our study (Table 6 and Table 7). Morocho et al. showed a COP of 3.5 s at 6 h post-resuscitation in adult patients with septic shock [12] and Schriger et al. proposed a normal upper limit for pediatric patients of 1.9 s [15]. In our study, the COPs obtained excellent PNV (>0.92) in all measurement times for both CRT and SL, highlighting the great utility of both tissue perfusion markers as screening tools. Similar results have been reported in a smaller observational study (*n* = 34) conducted by Shaker et al. in adult patients with abdominal sepsis [5].

SL turned out to be an independent risk factor for both outcomes except at IPO and 12 h measurements for 30-day mortality and ECMO requirement. We found an inversely proportional relationship between the measurement time and the COP. Likewise, Fuernau et al. reported an initial SL COP of 5 mmol and a COP of 3.1 mmol/L at 8 h in cardiogenic shock [16]. Scolari et al. reported a COP of 3.27 mmol/L at 6 h, 3.15 mmol/L at 12 h, and 1.55 mmol/L at 24 h [17]. A normal clearance of the SL is expected, increasing over time, and eventually decreasing the COP (Table 3). In a cohort study, failure to improve lactate levels after 24 h of mechanical circulatory support therapy was associated with 100% mortality [17].

The normal SL clearance is one of the possible reasons that explain why both tissue perfusion markers are not abnormal at the same time point and differ in the significance timing. Other possible causes include the factors that can influence SL such as muscle activity, medications, or medical conditions, especially in type B lactic acidosis, which is caused by an impairment of the body’s ability to clear lactate production [18]. In addition, differences in early measurements might be due to major fluid shifts, residual anesthesia, and hypothermia [12].

Even though the SL AUC showed a better discriminative capacity for both outcomes, the multivariate analysis suggests that the CRT has a better independent predictive capacity at IPO and 12 h, which has also been reported in some studies [5,12]. Likewise, in adult patients with septic shock, CRT-guided treatment has been shown to be non-inferior to serum lactate-guided treatment [13].

Our study is a starting point, in which the utility of the CRT is proven, and it may have the potential to suggest a different approach regarding the use of perfusion markers for monitoring pediatric CHS. Serum lactate remains the main marker of tissue perfusion in CHS; however, phlebotomy blood losses play a key role in anemia and the need for RBC transfusions [6], especially in prematurity, low weight-for-age children, and during the first weeks of life. Small preterm infants are often the most critically ill, require more blood tests, and suffer a greater loss of blood because they have a reduced circulating RBC volume [19,20,21]. This makes the development and validation of a non-invasive monitoring method vital for reducing the number of phlebotomies and their potential negative impact, particularly for the sickest patients. It is also important to limit testing to only the most critical cases [19]. CRT is a risk-free perfusion clinical marker, does not require blood samples or specific technology, and has immediate results [22]. A combined SL and CRT monitoring strategy for the studied population should be considered.

The subjective measure of CRT is controversial, and it can be unreliable [13]. However, objective CRT measurement using a chronometer demonstrated good interrater reliability [11,13]. We suggest implementing a strict protocol for measuring CRT, starting with adequate education on the correct technique, with the possibility of implementing tools such as the glass (Appendix A) and the stopwatch we used for the study. This will improve the precision of this clinical marker.

## 5. Strengths and Limitations

Among the strengths, this is the first study in which the predictive capacity of SL and CRT was measured for mortality and the ECMO requirement in patients undergoing pediatric CHS. Moreover, a strict and reproducible method was used for the measurement of the CRT. The research team was trained to apply the same measurement technique, using a specific instrument to apply pressure (Appendix A) and a stopwatch. For the evaluation of the CRT, integer values were established to increase its applicability in clinical practice.

## 6. Conclusions

Both CRT and SL were related to 30-day mortality and the need for postoperative ECMO in pediatric patients undergoing CHS. CRT is a reliable perfusion clinical marker and, given its advantages over other biomarkers that require blood samples, it should be included as a monitoring tool. A combined SL and CRT monitoring strategy will likely decrease the serial blood sample complication risks without negatively impacting the outcomes.

## Figures and Tables

**Figure 1 children-10-00875-f001:**
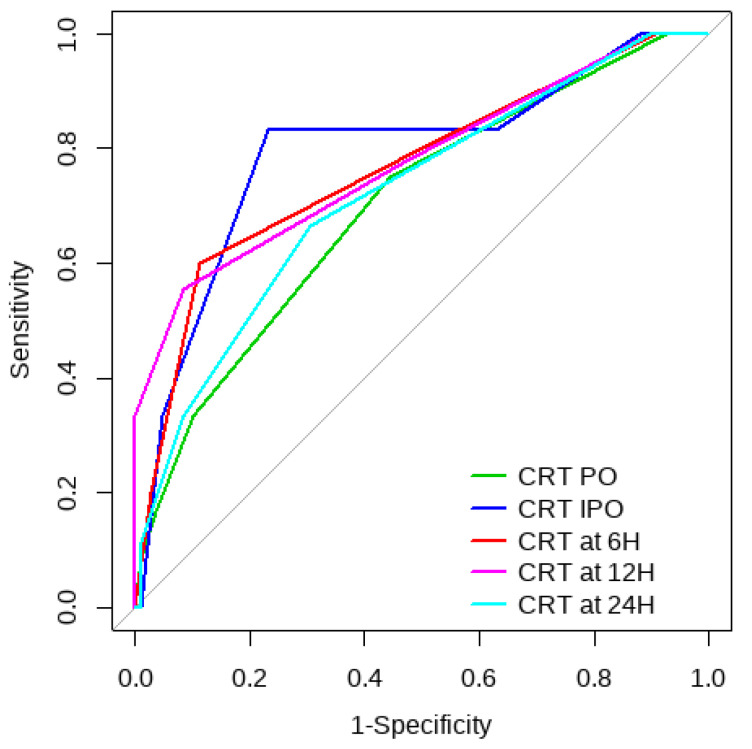
CRT ROC curve for mortality risk prediction.

**Figure 2 children-10-00875-f002:**
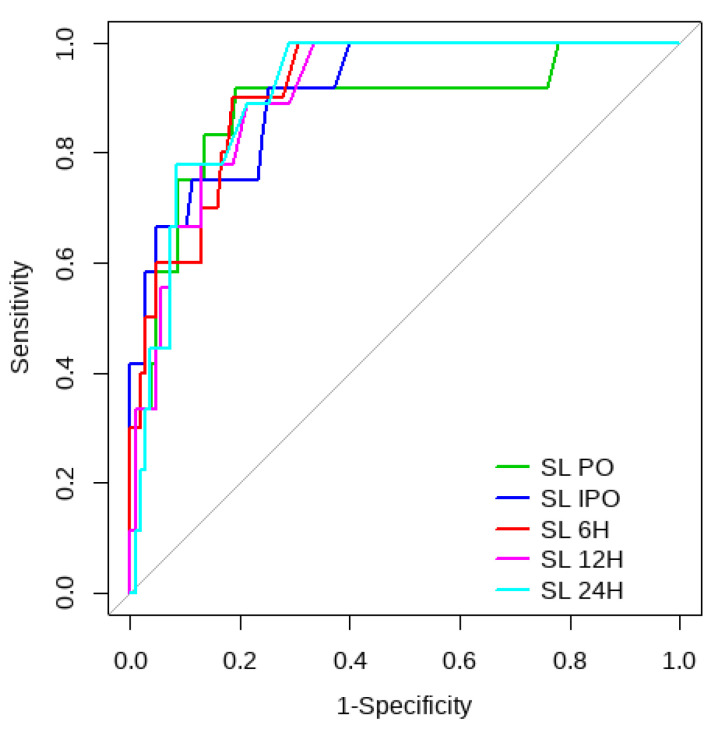
SL ROC curve for mortality risk prediction.

**Figure 3 children-10-00875-f003:**
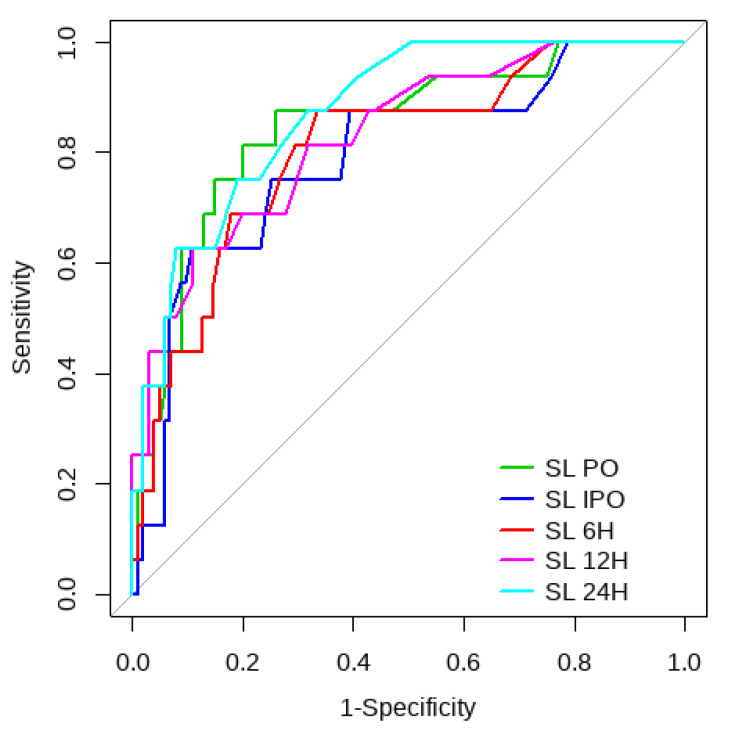
CRT ROC curve to predict postoperative ECMO requirement risk.

**Figure 4 children-10-00875-f004:**
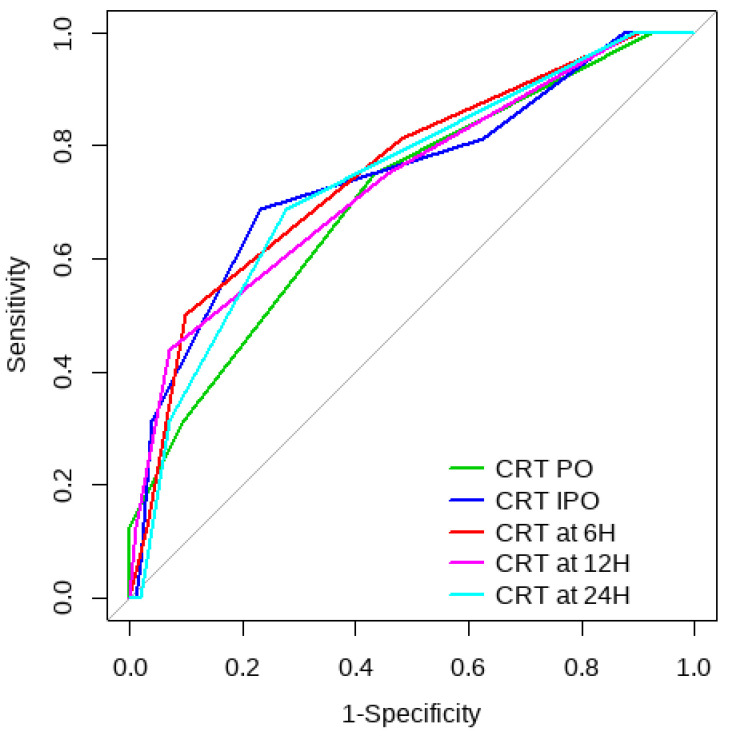
SL ROC curve to predict postoperative ECMO requirement risk.

**Table 1 children-10-00875-t001:** Sociodemographic information and clinical characteristics of the participants.

Variable	*n*	*n* = 120
Age in months	120	
Age in months		4.5 (0.8, 11.0)
Newborns (<1 month)		30 (25.0%)
Infants (1–12 months)		65 (54.2%)
Toddler (1–5 years)		25 (20.8%)
Sex	120	
Female		60 (50.0%)
Male		60 (50.0%)
Weight (kg)	120	5 (3.5, 7.1)
Height (cm)	120	60 (50.0, 69.0)
Surgery condition	120	
Elective		57 (47.5%)
Urgent/Emergency		63 (52.5%)
CPB time (min)		79 (55.0, 125.5)
Aortic clamp time (min)		58 (29.2, 78.8)
Medical history	120	
Pulmonary disease		33 (27.5%)
Neurologic disease		32 (26.7%)
Renal disease		11 (9.2%)
Hepatic disease		7 (5.9%)
Endocrine disease		14 (11.8%)
Low weight-for-age		62 (52.0%)
Low weight-for-height		42 (35.0%)
Preterm		27 (22.5%)
Chromosomes anomalies	120	
Down syndrome		19 (15.8%)
Other major chromosome anomalies		14 (11.7%)
Preoperative laboratories		
Hematocrit	119	38 (34.0, 45.4)
Platelet count	119	305,000.0 (230,000.0, 395,500.0)
INR	111	1.1 (1.0, 1.2)
Creatinine	113	0.3 (0.3, 0.4)
Glycemia	101	91 (84.0, 107.0)
Outcomes	120	
30-day mortality		12 (10.0%)
Postoperative ECMO		16 (13.3%)

ECMO: extracorporeal membrane oxygenation, INR: international normalized ratio.

**Table 2 children-10-00875-t002:** RACHS-1 and 30-day mortality of the procedures.

RACHS-1	Total *n* = 120	30-day Mortality *n* = 12	POP ECMO *n* = 16
Not classifiable	5 (4.16%)	0 (0.00%)	1 (6.25%)
1	20 (16.66%)	1 (8.33%)	2 (12.5%)
2	33 (27.50%)	0 (0.00%)	3 (18.75%)
3	33 (27.50%)	2 (16.66%)	2 (12.50%)
4	15 (12.50%)	1 (8.33%)	3 (18.75%)
5–6	14 (11.66%)	8 (66.66%)	5 (31.25%)

RACHS-1: risk adjustment for congenital cardiovascular surgeries, POP ECMO: postoperative extracorporeal membrane oxygenation requirement.

**Table 3 children-10-00875-t003:** Median and Interquartile range of CRP and SL measurements.

Variable	*n* = 120	
MT	CRP-Seconds	SL-(mmol/L)
PO	2.0 (2.0, 3.0)	2.1 (1.6, 3.3)
IPO	3.0 (2.0, 4.0)	2.1 (1.5, 3.7)
6 h	3.0 (2.0, 3.0)	1.7 (1.1, 2.8)
12 h	2.0 (2.0, 3.0)	1.2 (1.0, 2.0)
24 h	2.0 (2.0, 3.0)	1.1 (0.8, 1.7)

MT: measurement time, PO: Preoperative, IPO: Immediate Postoperative, CRT: Capillary Refill time, SL: Serum lactate.

**Table 4 children-10-00875-t004:** CRT and SL relationship with 30-day mortality in the univariate and multivariate analysis.

Variable	Univariate Analysis			Multivariate Analysis		
	OR	CI 95%	*p*-Value	OR	CI 95%	*p*-Value
PO						
CRT	2.00	(1.19–3.45)	0.009	1.42	(0.75–2.72)	0.278
SL	1.40	(1.21–1.69)	0.000	1.43	(1.17–1.81)	0.000
IPO						
CRT	1.93	(1.25–3.36)	0.002	1.37	(1.17–1.64)	0.000
SL	1.47	(1.27–1.79)	0.000	2.34	(0.92–6.85)	0.074
6 h						
CRT	2.98	(1.54–6.27)	0.001	1.71	(1.02–2.89)	0.043
SL	1.51	(1.26–1.89)	0.000	1.20	(1.00–1.48)	0.044
12 h						
CRT	4.31	(1.98–11.19)	0.000	1.45	(1.24–1.76)	0.000
SL	1.34	(1.16–1.62)	0.000	1.47	(0.88–2.32)	0.118
24 h						
CRT	1.57	(1.01–2.68)	0.047	1.59	(0.71–3.60)	0.255
SL	1.30	(1.07–1.77)	0.009	1.25	(1.02–1.69)	0.029

PO: Preoperative, IPO: Immediate Postoperative, CRT: Capillary Refill time, SL: Serum lactate, OR: Odds Ratio, CI: Confidence Interval.

**Table 5 children-10-00875-t005:** CRT and SL relationship with a postoperative ECMO requirement in the univariate and multivariate analysis.

Variable	Univariate Analysis			Multivariate Analysis		
	OR	CI 95%	*p*-Value	OR	CI 95%	*p*-Value
PO						
CRT	2.2	(1.37–3.79)	0.001	1.75	(0.98–3.25)	0.057
SL	1.35	(1.17–1.60)	0.000	1.22	(1.03–1.47)	0.018
IPO						
CRT	1.81	(1.22–2.96)	0.003	1.30	(1.13–1.53)	0.000
SL	1.22	(1.09–1.39)	0.001	1.56	(0.75–3.38)	0.232
6 h						
CRT	2.74	(1.56–5.16)	0.000	1.55	(1.04–2.42)	0.031
SL	1.32	(1.14–1.57)	0.000	1.32	(1.05–1.67)	0.003
12 h						
CRT	3.03	(1.65–6.13)	0.000	1.19	(1.04–1.35)	0.009
SL	1.42	(1.20–1.78)	0.000	1.39	(0.82–2.13)	0.179
24 h						
CRT	1.63	(1.08–2.88)	0.020	1.96	(1.05–3.80)	0.034
SL	2.04	(1.42–3.26)	0.000	1.87	(1.33–2.91)	0.000

PO: Preoperative, IPO: Immediate Postoperative, CRT: Capillary Refill time, SL: Serum lactate, OR: Odds Ratio, CI: Confidence Interval.

**Table 6 children-10-00875-t006:** AUC and optimal COP of CRT and SL to predict 30-day mortality.

Variable	AUC (95% CI)	COP	COP Sensitivity (%, 95% CI)	COP Specificity (%, 95% CI)	PPV (95% CI)	PNV (95% CI)
CRT PO	0.701 (0.55, 0.852)	3	0.75 (0.43, 0.94)	0.56 (0.46, 0.65)	0.16 (0.11, 0.52)	0.95 (0.83, 0.97)
CRT IPO	0.796 (0.643, 0.948)	4	0.83 (0.52, 0.98)	0.77 (0.68, 0.84)	0.29 (0.20, 0.79)	0.97 (0.90, 0.98)
CRT 6 h	0.768 (0.596, 0.939)	4	0.60 (0.26, 0.88)	0.89 (0.81, 0.94)	0.33 (0.21, 0.71)	0.96 (0.85, 0.98)
CRT 12 h	0.777 (0.588, 0.965)	4	0.56 (0.21, 0.86)	0.92 (0.85, 0.96)	0.36 (0.22, 0.74)	0.96 (0.84, 0.98)
CRT 24 h	0.723 (0.552, 0.894)	3	0.67 (0.30, 0,92)	0.69 (0.60, 0.78)	0.15 (0.11, 0.53)	0.96 (0.84, 0.97)
SL PO	0.880 (0.756, 1.004)	3.2	0.92 (0.61, 1.00)	0.81 (0.72, 0.88)	0.35 (0.25, 0.96)	0.99 (0.92, 0.99)
SL IPO	0.910 (0.833, 0.988)	3.1	0.92 (0.61, 1.00)	0.75 (0.66, 0.83)	0.29 (0.21, 0.95)	0.99 (0.92, 0.99)
SL 6 h	0.914 (0.845, 0.984)	3	0.90 (0.55, 1.00)	0.81 (0.73, 0.88)	0.31 (0.21, 0.95)	0.99 (0.92, 0.99)
SL 12 h	0.907 (0.831, 0.983)	2	0.89 (0.52, 1.00)	0.79 (0.70, 0.86)	0.26 (0.18, 0.94)	0.99 (0.92, 0.99)
SL 24 h	0.913 (0.845, 0.981)	1.5	1.00 (0.66, NA)	0.71 (0.62, 0.80)	0.22 (0.16, NA)	1.00 (0.94, 1.00)

AUC: Area under the curve, COP: cut-off point, CI: Confidence interval, PPV: Predictive positive value, PNV: Predictive negative value, CRT: capillary refill time, PO: Preoperative, IPO: immediate postoperative.

**Table 7 children-10-00875-t007:** AUC and optimal COP of CRT and SL to predict postoperative ECMO requirement.

Variable	AUC (95% CI)	COP	COP Sensitivity (%, 95% CI)	COP Specificity (%, 95% CI)	PPV (95% CI)	PNV (95% CI)
CRT PO	0.706 (0.575, 0.838)	3	0.75 (0.48, 0.93)	0.57 (0.47, 0.66)	0.21 (0.15, 0.53)	0.94 (0.82, 0.96)
CRT IPO	0.747 (0.606, 0.888)	4	0.69 (0.41, 0.89)	0.77 (0.68, 0.85)	0.31 (0.22, 0.63)	0.94 (0.84, 0.96)
CRT 6 h	0.754 (0.626, 0.882)	4	0.50 (0.25, 0.75)	0.90 (0.83, 0.95)	0.44 (0.29, 0.71)	0.92 (0.79, 0.96)
CRT 12 h	0.737 (0.601, 0.873)	4	0.44 (0.20, 0.70)	0.93 (0.86, 0.97)	0.50 (0.32, 0.75)	0.91 (0.77, 0.96)
CRT 24 h	0.739 (0.616, 0.861)	3	0.69 (0.41, 0.89)	0.72 (0.62, 0.81)	0.28 (0.20, 0.59)	0.94 (0.82, 0.96)
SL PO	0.845 (0.737, 0.953)	2.8	0.87 (0.62, 0.98)	0.74 (0.64, 0.82)	0.35 (0.25, 0.83)	0.97 (0.89, 0.98)
SL IPO	0.791 (0.664, 0.918)	5	0.62 (0.35, 0.85)	0.89 (0.82, 0.95)	0.47 (0.33, 0.75)	0.94 (0.84, 0.97)
SL 6 h	0.808 (0.692, 0.924)	2	0.87 (0.62, 0.98)	0.67 (0.57, 0.76)	0.29 (0.21, 0.79)	0.97 (0.89, 0.98)
SL 12 h	0.830 (0.720, 0.940)	2.8	0.62 (0.35, 0.85)	0.89 (0.81, 0.94)	0.48 (0.33, 0.75)	0.93 (0.83, 0.97)
SL 24 h	0.873 (0.793, 0.953)	1.7	0.75 (0.48, 0.93)	0.81 (0.72, 0.88)	0.39 (0.27, 0.73)	0.95 (0.86, 0.97)

AUC: Area under the curve, COP: cut-off point, CI: Confidence interval, PPV: Predictive positive value, PNV: Predictive negative value, CRT: capillary refill time, PO: Preoperative, IPO: immediate postoperative.

## Data Availability

Data sharing not applicable.

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
