# Peer review of "Capillary Refill Time and Serum Lactate as Predictors of Mortality and Postoperative Extracorporeal Membrane Oxygenation Requirement in Congenital Heart Surgery"

_children, 2023, doi:10.3390/children10050875_

Round 1

Reviewer 1 Report

General comments

The utilization of various tissue perfusion biomarkers is crucial for guiding treatment decisions in critically ill pediatric patients. Despite serum lactate (SL) being recognized as the preferred biomarker, its drawbacks, such as the need for blood samples that increase the risk of anemia and infection, have been highlighted. As a non-invasive and easily reproducible alternative, capillary refill time (CRT) has been employed.

In their paper, Cruz G et al investigate the use of CRT as a predictor of 30-day mortality and extracorporeal membrane oxygenation requirement in pediatric patients undergoing cardiac surgery, in comparison with serum lactate. The authors propose a reproducible method for measuring CRT, providing insights into its potential as a reliable non-invasive predictor for short-term worst outcome after cardiac surgery in pediatric patients.

Comments to the authors

Congratulations to Cruz G and colleagues for their thorough analysis and aligned conclusions based on the analyzed sample size. However, there are certain issues that require attention:

Major comments.

1) In order to perform a valid multivariate linear or logistic regression analysis, several assumptions must be met, including an event-to-variable ratio > 8-10. Due to the low number of events (e.g. N = 12 for 30-day mortality), there is a risk of creating an overfitted model, which could influence the final results. Furthermore, since there is likely to be a high correlation between capillary refill time (CRT) and serum lactate (SL), the authors should consider the risk of multicollinearity and analyze this issue (e.g. by performing a Pearson correlation coefficient). 

As a matter of facts, although both CRT and SL are significant in the univariate analysis, the significance (p-value) of the two dependent variables behaves randomly in the multivariate analysis, independent of the variable of interest. Therefore, to overcome this issue, the authors should either maintain univariate analysis or perform a covariate adjustment using propensity score, which has been demonstrated to perform better in small populations than multivariate regression analysis alone.

2) Authors stated that: “Logistic regression with Firth's penalty was chosen, with the aim of predicting mortality and the requirement of ECMO based on the available markers. A univariate and multivariate regression analysis was performed including SL and CRT for each measurement time”.

It's important to note that when using Firth's penalty with continuous covariates, the technique can affect the interpretation of regression coefficients since the estimated coefficients may differ from those obtained with standard logistic regression. Therefore, the authors should provide the results from standard logistic regression to determine the potential bias given by this kind of data manipulation.

3) CRT and SL levels may be affected by vasopressor therapy. Therefore, the authors should provide a new table that represents the inotropic and vasopressor therapy immediately after the procedure and possibly at the timepoint when the CRT and SL were acquired. Additionally, the authors should clarify this data by creating another column for patients that had the worst outcome (i.e. 30-day mortality and ECMO requirement – as for Table 2).

Minor comments.

1) To better appreciate the distribution of CRT and SL levels in patients with worst outcomes, I would suggest that the authors edit Table 3 by adding the column “Mortality” as in Table 2.

2) The timepoint (30-day) for the mortality outcome should be clarified throughout the manuscript, as it is currently only mentioned in the “Statistical method” section and Table 1.

3) While the tables are concise and clear in many aspects, the graphic quality could be improved as follows: 

            - In Table 1 and 3 there is no need to clarify at the bottom of the tables that the continuous variables are represented as median (IQR), as it is already mentioned in the “Statistical method” section.

            - Table 1: Other mayor chromosome anomalies 14 patients does not correspond to the 14.1% of the entire population.

            - Uniform the p-value label between Table 4 and 5.

Reviewer 2 Report

This article introduces the feasibility of CRT, a very simple indicator used to predict mortality and ECMO requirement in congenital heart surgery, which has some value and significance. The main problem of the article is the CRT measurement. They adopts the manual compression and stopwatch timing, which is subjective and can not be accurate. This measurement remains to be further verified by other medical institutions and physicians. The other concerns are listed below.

1. Language needs some editing, for example, line 33-34 ”Both tissue perfusion markers resulted in mortality and extracorporeal oxygenation requirement predictors”; line 35 both markers; Line 35 this population. What are the markers and this population? Language use problems make the manuscript a bit hard understanding.

2. Presentation of Tables or figures needs to be modified, which should follow the corresponding text immediately, for better readability.

3. Texts does not match Tables, for example, Line 153, median age in the text is 5 months while in the table was 4.5 months.; line l54, the most common age group was infants(54%). Please double check for all the manuscript.  

4. Line 156, 22,5 should be 22.5, a Typing mistake; please define premature, whats the meaning of premature

5. Please describe table 2, 3, 7, and Figure 1-4, what do they mean and what do the authors want to convey to the readers by using them?

6. Line 108, please define RACHS when it firstly appears.

7. Line 160, what does the mean of greater? greater predictive capacity than that of other time points or than that of SL?

8. Line 173, please define COP when it firstly appears.

Author Response

Please see atachment

Round 2

Reviewer 1 Report

The authors have been carefully reviewed every point raised during the first round of review, addressing them in a proper way. Authors’ thorough revisions have significantly improved the quality of the manuscript, and it now meets the standards required for publication. I appreciate the time and effort invested in responding to the reviewers' comments and suggestions.

Reviewer 2 Report

The authors addressed most of my concerns. I have no further comments. Thanks.